# Soil Rehabilitation Promotes Resilient Microbiome with Enriched Keystone Taxa than Agricultural Infestation in Barren Soils on the Loess Plateau

**DOI:** 10.3390/biology10121261

**Published:** 2021-12-02

**Authors:** Dong Liu, Parag Bhople, Katharina Maria Keiblinger, Baorong Wang, Shaoshan An, Nan Yang, Caspar C. C. Chater, Fuqiang Yu

**Affiliations:** 1State Key Laboratory of Soil Erosion and Dryland Farming on the Loess Plateau, Institute of Soil and Water Conservation, Chinese Academy of Sciences and Ministry of Water Resources, Northwest A&F University, Xianyang 712100, China; wangbaorong92@163.com (B.W.); shan@ms.iswc.ac.cn (S.A.); 2The Germplasm Bank of Wild Species, Yunnan Key Laboratory for Fungal Diversity and Green Development, Kunming Institute of Botany, Chinese Academy of Sciences, Kunming 650201, China; 3Department of Biological Sciences, Faculty of Science and Engineering, University of Limerick, V94 T9PX Limerick, Ireland; paragbhoplebt@gmail.com; 4Department of Forest and Soil Sciences, Institute of Soil Research, University of Natural Resources and Life-Sciences, 1190 Vienna, Austria; katharina.keiblinger@boku.ac.at; 5College of Biology and the Environment, Nanjing Forestry University, Nanjing 210037, China; nyang@njfu.edu.cn; 6Royal Botanic Gardens, Kew, Richmond TW9 3AE, UK; caspar.chater@gmail.com

**Keywords:** Loess soils, keystone taxa, rehabilitated lands, agricultural soils, microbial network

## Abstract

**Simple Summary:**

Soil degradation accompanied by agricultural intensification is threatening the existence of dryland areas globally. However, improvement in soil quality in these areas through plant restoration has been greatly successful. Under this practice, soil microorganisms, especially keystone taxa, exert positive feedback on soil bio-functionality by recycling nutrients. Nevertheless, to date, it remains unclear as to how similar are the changes in these keystone taxa in dryland barren soils and agricultural soils, with comparison perspective. The earlier longer-term experiments in Loess soils have shown profound positive effects of plant restoration and establishment of more complex soil microbial networks in the presence of higher nutrient accumulation. The current work shows more keystone taxa that tend to exist in rehabilitated conditions when compared to their agricultural soil counterparts.

**Abstract:**

Drylands provide crucial ecosystem and economic services across the globe. In barren drylands, keystone taxa drive microbial structure and functioning in soil environments. In the current study, the Chinese Loess plateau’s agricultural (AL) and twenty-year-old rehabilitated lands (RL) provided a unique opportunity to investigate land-use-mediated effects on barren soil keystone bacterial and fungal taxa. Therefore, soils from eighteen sites were collected for metagenomic sequencing of bacteria specific 16S rRNA and fungi specific ITS2 regions, respectively, and to conduct molecular ecological networks and construct microbial OTU-based correlation matrices. In RL soils we found a more complex bacterial network represented by a higher number of nodes and links, with a link percentage of 77%, and a lower number of nodes and links for OTU-based fungal networks compared to the AL soils. A higher number of keystone taxa was observed in the RL (66) than in the AL (49) soils, and microbial network connectivity was positively influenced by soil total nitrogen and microbial biomass carbon contents. Our results indicate that plant restoration and the reduced human interventions in RL soils could guide the development of a better-connected microbial network and ensure sufficient nutrient circulation in barren soils on the Loess plateau.

## 1. Introduction

Nearly 40% of the global terrestrial surface is covered by drylands which conscript a range of benefits that ecosystems cater to societies across the globe [1]. However, on the planetary scale, soil degradation accompanied by agricultural intensification is threatening the existence of drylands [2]. Moreover, severe soil degradation has caused dramatic loss of ecosystem services such as water filtration, wind prevention, and sand fixation, etc., thereby further accelerating soil degradation via erosion and poor vegetation cover [3,4,5]. To support sustainable development, soil rehabilitation of eroded land has attracted considerable interest worldwide [6,7]. Ecological restoration (revegetation for plant community development) is a largely accepted soil rehabilitation strategy due to its positive effects on soil physical structure, soil organic matter (SOM) accumulation, and so optimizing the existent ecosystem services [8,9]. Land rehabilitation includes vegetation recovery in barren soils and land-use transformation from crop lands to artificial forests, grasslands, or shrub lands [10,11,12]. In China, since 1999, over seven million hectares of cultivated lands have been rehabilitated with improvements in overall soil qualities including nutrients (i.e., available phosphorus, soil nitrogen, and soil organic carbon), enzyme activities, as well as microbial biomass [13,14,15,16]. These interventions have further contributed to enhancing the diversity and community structure of the soils’ microbial communities.

Apparently, changes in land use can also induce changes in microbial phyla. For instance, some Proteobacteria are copiotrophic, and their incidence correlates with increased soil C and N contents after revegetation [17,18,19]. On the other hand, 30 years of land restoration in the Loess plateau showed a significant effect on soil properties and soil nutrient cycle [13]. Agricultural management practices such as stubble retention or reduced tillage lead to retention of cellulose and chitin-like residues in soils which are preferentially decomposed more efficiently primarily by fungal members belonging to Chytridiomycota [20,21,22] in comparison to other fungi such as the Agaricomycetes and or Pezizomycetes from Ascomycota and Basidiomycota fungal phyla. Soil microbiomes are fundamental to successful ecological restoration due to their ability to regulate nutrient cycling and maintain biogeochemical processes [23,24], which in turn are potential indicators of soil bio-functioning. Keystone taxa affect many microbial associations which further drive changes in soil microenvironment along with other physicochemical factors governing the same [25,26,27]. Moreover, keystone taxa highlight any shifts in microbial community structure and the associated compositional turnover [28].

Network analysis of microbe-microbe interactions can reveal details in microbial community complexities and identify ‘keystone taxa‘ which drive community composition and function, irrespective of their abundance in the soil matrix. Several studies conclude that changes in land use affect changes in soil microbial communities and their respective keystone taxa [29,30]. For example, the keystone microbes significantly increased in vegetation-restored soils in comparison to the degraded bare lands [29], portraying an effective strategy for the restoration of degraded croplands. Likewise, more stable and complex microbial networks with greater number of keystone taxa were detected in the forest ecosystems when compared to the arable soils in the Loess region. Moreover, these keystone taxa have versatile functions in C and N elemental cycling and perform key roles in sustainable ecological systems [26]. However, similar studies for dryland systems are limited, and very few studies to date have quantified dryland microbial keystone species in a network guild in combination with comprehensive soil microbiome analyses using high-throughput metagenomic DNA sequencing approach.

The arid to semi-arid Chinese Loess Plateau (CLP) provides a unique study location for longer term soil restoration in the field and can inform management practices across global drylands that form the largest terrestrial biome on earth. In this regard, coupled with increasing temperature, climate change will increase aridity, which in turn is envisioned to result in a decline or suppression of microbial abundance and activity in soils [31,32]. Therefore, although China’s Grain for Green (GFG) initiative was initially implemented to address pressing environmental problems such as land restoration in China, it has over the years garnered crucial global implications. With the development of the GFG, land use structure on the CLP has been converted drastically [33,34] and the majority of CLP land cover has been designated as rehabilitated lands (RL) or as agricultural lands (AL). The present study avails the benefits of the GFG initiative to study the effects of changes in land management practices on soil microbiomes and the keystone taxa in soils across dryland landscapes that have been restored and managed over the past twenty years.

Soils of the Chinese Loess Plateau have been part of several assessments [3,4,5] and long-term studies [14,29] that investigated the effects of anthropogenic interference on ecosystem properties. Nonetheless, simultaneous exploration of both the soil microbiome, and the integral keystone taxa to reflect microbial network properties is still limited, which could be a great resource for future studies. The soil properties and microbial communities are interlinked [35,36] and therefore in our H1 hypothesis we expect significant variations in the microbial diversity and community composition between the RL and the AL land-use type soils. Carbon through differential input land, and likewise the other important nutrients such as the N, P, S are accumulated differently in the diverse vegetative habitats/land-use types. In the AL land-use type, soil microbes have substrate-competitive pressure due to constantly anthropogenic nutrients inputs [37]. Whereas, in RL, without artificial disturbances, soil nutrients accumulate more naturally and there is higher C input in the RL soils as generally seen in the forest ecosystems [27,38] and hence the microbial-mediated nutrient build-up, especially for bacterial microbiota, as soil bacteria are regarded as more important mediators for rapid nutrient cycle [39]. Moreover, during forest ecosystem recovery, bacterial community interactions were complex whereas the fungal networks were seen to be relatively simpler and more isolated [40,41]. Based on these understandings, in our second hypothesis (H2) we expect a more complex and cooperative bacterial network in rehabilitated land (RL), along with more keystone taxa and richer functional modules.

## 2. Materials and Methods

### 2.1. Site Description

In the northern part, the Grain for Green (GFG) initiative led to the development of various land-use patterns. For instance, in the Ziwuling forest belt of the Ansai county and the Liandaowan town, many previously uncultivable hilly slope farmlands are being revegetated as grasslands, shrub lands, and forest lands [38,42,43,44] forming a cluster of rehabilitated lands (RL). Consequently, immense improvement in the soil quality was noted in these RL clusters. In the southern part of the Loess plateau such as the Weibei upland, Guanzhong plain and Luochuan tableland, wide-spread agricultural lands (AL) (34°18′–35°50′ N; Appendix A) prominently grow *Malus pumila*, *Zea mays*, *Amygdalus persica* and *Pyrus* spp. [45,46]. Large areas of *M. pumila* have been cultivated for a period of approximately twenty years with high crop turnover. Beside *M. pumila*, *A. persica* and *P*. spp. were the other two local economically important plants [45,46]. *Z. mays* is usually cultivated as the main crop. The climate is arid and semi-arid with low precipitation and agriculturally used soils facing high evaporation rates. These soils are moderate saline-alkali and characterized by a high pH (7.7–8.5) and salt content (averaged 514; Appendix A). The accustomed fertilization practices of local farmers include the application of organic fertilizer and high-efficiency compound fertilizer (three nutrients compound N, P, K) fertilizer [47,48,49]. In compliance with the GFG (year 1999 to 2019) in the Loess hilly regions [42], large areas of slopes or hilly farmlands have been re-vegetated to form rehabilitated land (RL) areas since 1999, and tree species such as *Pinus tabuliformis*, *Betula platyphylla*, *Artemisia gmelinii*, *Caragana Korshinskii*, *Robinia pseudoacacia*, *Bothriochloa* spp., *Artemisia scoparia*, *Stipa grandis* have been grown. At both the AL (agricultural land) and the RL (rehabilitated land), nine sites were selected at the landscape scale in the Chinese Loess Plateau. All soils were collected during the non-fertilizer application period from June to July 2018. To minimize spatial heterogeneity and capture soil variation, at each site six soil cores were used in each of the three subplots (20 × 20 m, distance between the subplot is 100 m), to randomly collect soil (using a drill size of 20 cm length and 2 cm diameter). Soils from the cores were bulked and sieved using <2 mm mesh to obtain a composite sample of >100 g. From this, a 10 g subsample was immediately wrapped in aluminum foil, quenched with liquid N_2_, and stored at −80 °C prior to extraction of soil metagenomic DNA. Microbial biomass was analyzed from the aliquots of samples stored at 4 °C. The remaining aliquots were air-dried for subsequent analyses of soil physicochemical parameters.

### 2.2. DNA Extraction and PCR Amplification

Genomic DNA was extracted from 0.5 g soil using commercial kits (MoBioPower, Carlsbad, CA, USA) and stored at −20 °C prior to PCR amplification according to Liu et al., (2018b) [50]. For bacteria, the 16S rRNA gene V4 hypervariable region was amplified using the 520F (AYTGGGYDTAAAGNG) and 802R (TACNVGGGTATCTAATCC) primer pair, respectively. The fungal internal transcribed spacer 2 (ITS 2) region was amplified using a mix of ITS3 and ITS4 specific primer sets, according to Fujita et al., (2001) [51]. After PCR thermal cycling, amplicons were extracted from 2% agarose gels and purified using the commercial AP-GX-500 DNA gel extraction kit, with a final quantification on a Microplate reader (BioTek, FLX800, Winooski, VT, USA) using the dsDNA Assay Kit, (Invitrogen, P7589, Waltham, MA, USA).

### 2.3. Sequencing on Illumina MiSeq Platform

16S rRNA and ITS sequence libraries were constructed with the TruSeq Nano DNA LT Library Prep Kit and sequenced on an Illumina MiSeq platform, generating 300 bp paired-end reads (Personalbio Co., Ltd., Shanghai, China). Raw sequencing reads with low-quality bases and adapter contamination were trimmed with cut-adapt (v2.3) [52] and Vsearch (v2.13.4_linux_x86_64) software [53]. Rarefaction method was used to normalize the reads, which randomly selects the smallest sample size (49,729 and 60,856 for bacterial 16S rRNA and fungal ITS, respectively) from each sample to reach a uniform sequencing depth to predict the observed OTUs and their relative abundance at threshold sequencing depth [54]. High quality sequences (after sequence filtering, denoising, merging, and the removal of chimeric and singleton) were clustered into operational taxonomic units (OTUs) with 97% similarity using USEARCH OTU clustering (http://www.drive5.com/usearch/, accessed on 15 Februray 2020) [55]. For bacterial OTUs, UCLUST was used to assign taxonomy against the Greengenes database [56]. For fungal OTUs, BLAST was used to search reads against the UNITE database [57]. The taxonomic cutoff was set at generic level and OTUs assigned to the same classification level were grouped together based on their taxonomic affiliations. Raw sequence data were deposited in the NCBI Sequence Read Archive under accession numbers SRP 126,991 (Fungal sequences) and SRP 126,984 (Bacterial sequences).

### 2.4. Network and Keystone Taxa Analyses

Microbial networks of the Loess agricultural lands (AL) and rehabilitated lands (RL) were analyzed individually from the pipeline of molecular ecological network analysis (MENA) (http://ieg4.rccc.ou.edu/mena/help.cgi, accessed on 20 April 2021). Firstly, the original bacterial and fungal OTU tables were standardized and then pairwise Pearson correlation was performed between microbial OTUs and filtered with the criteria of *p* value < 0.01 and a coefficient > |0.7| [58]. OTU correlation matrices were then constructed based on random matrix theory [59,60]. In order to reflect the complexity and interactive status of microbial taxa within a network, the number of nodes (OTUs), the links (significant associations among OTUs), and the positive links (referring to cooperative relationships among OTUs) were analyzed from the MENA pipeline [59,60]. To reveal a network’s resistance to the external environment, network-level topological parameters including average clustering coefficient (avgCC), average connectivity (avgK), and the numbers of module and geodesic distance (GD) between nodes [60,61,62] were calculated. Node-level indices used for keystone taxa identification were: (i) among-module connectivity (Pi) used to indicate nodes as module connectors (Pi > 0.62); (ii) within-module connectivity (Zi), referring to highly connected nodes within modules as module hubs (Zi > 2.5); and (iii) important nodes to both the network and their own module coherence as network hubs (Zi > 2.5, Pi > 0.62). The nodes with either a high value of Zi or Pi were defined as keystone taxa including network hubs, module hubs, and connectors [63]. Detailed keystone information concerning keystone number, their assigned phyla, association to a functional module, and keystone taxa distribution features were sorted and visualized using OriginPro 8 software.

### 2.5. The Relationship between Network Topology and Environmental Variables

The importance of environmental variables on network topology was examined as described by Deng et al., (2012) [43]. The OTUs’ significance was defined as the square of Pearson correlation coefficients between relative abundance of OTUs in a network and environmental variables [64]. This was achieved by selecting a dataset (i.e., the fungal network of the agricultural lands with the number of 453 OTUs) to first calculate OTU significance, followed by MENA database submission, and then uploading the environmental trait file (13 variables in our case). The options of “correlation method: Pearson correlation coefficient” and “standardization method: standardize environmental data only (scale each factor to zero mean and unit variance)” were selected. The OTU-significance matrix was obtained with specific OTUs distributed in rows and environmental traits in columns. The final matrix table can be downloaded from the MENA pipeline. For the overall OTU-significance matrix, we chose specific options (Distance method: Euclidean distance) and consequently ran a Mantel test on each of these environmental factors.

### 2.6. Statistical Analysis

The alpha diversity indices for bacterial and fungal communities of Loess agricultural lands (AL) and rehabilitated lands (RL) were calculated based on the normalized microbial OTUs table. Estimated OTU richness was evaluated using the Chao1 index, and the Shannon index was used to consider both the number and abundance of OTUs, while the Simpson index was used to consider the number and the relative abundance of individual OTUs. Significant differences between the AL and RL soils were tested using two independent sample T-tests. Beta-diversity of the soil microbial communities was determined using the UniFrac metric [65] in the MOTHUR [66] platform. Unweighted Non-Metric Multi-Dimensional Scaling (NMDS) plot analysis was performed using the vegan package [67] of R software and was based on the unweighted UniFrac distance matrix. Furthermore, ANOSIM (analysis of similarities) was conducted to test differences in bacterial or fungal community structure between AL and RL soils. Dispersion analysis (Permdisp) was used to confirm the grouping result (between AL and RL), and test whether the grouping in NMDS was due to significant differences among group centroids or heteroscedasticity between groups. Variation Partitioning Analysis (VPA) was performed to explore the proportion of variability in microbial communities that can be explained by land-use types, and edaphic and spatial variables (as evaluated by geographic distance between sampling sites). Linear discriminant analysis (LDA) effect size was used to investigate microbial biomarkers across the studied agricultural and rehabilitated soils. The most obvious biomarkers were selected based on a threshold of LDA score > 2.0 and *p*-value < 0.05. PICRUSt (phylogenetic investigation of communities by reconstruction of unobserved states) was used to predict functional abundances based on the 16S rRNA gene sequencing data [68].

## 3. Results

### 3.1. Variations in Edaphic Properties and Microbial Communities across Different Land-Use Types

All the investigated rehabilitated lands (RL) are situated at higher altitude (1271 ± 129 m a.s.l.) as compared to the agricultural lands (AL) (798 ± 377 m). Soil organic carbon (SOC), total nitrogen (TN), and total phosphorus (TP) contents were all higher in AL (SOC, 4.7 ± 0.8 g kg^−1^; TN, 2.3 ± 0.2 g kg^−1^; TP, 1.1 ± 0.3 g kg^−1^) than in RL (SOC, 3.6 ± 1.4 g kg^−1^; TN, 1.6 ± 0.9 g kg^−1^; TP, 0.5 ± 0.04 g kg^−1^). However, soil pH, dissolved organic carbon (DOC), and microbial biomass carbon (MBC) contents were higher in RL (pH, 8.9 ± 0.2; DOC, 73 ± 36 mg kg^−1^; MBC, 243 ± 192 mg kg^−1^) than in AL (pH, 8.3 ± 0.3; DOC, 56 ± 16 mg kg^−1^; MBC, 151 ± 41 mg kg^−1^) (Appendix A).

Variation partition analysis showed that the total microbial community variation was mainly explained by land-use type (Bacteria, 27.7%; Fungi, 30.2%), followed by edaphic variables (Bacteria, 14.2%; Fungi, 17.9%), and spatial variables (8% for both bacteria and fungi) (Figure 1). Land use and edaphic properties together explained 12.6% (bacterial) and 10.4% (fungal) of the community variations. Taken together, despite the various unknown (unexplained percentage ~60%) factors that influence community variation, soil properties derived by land use significantly shape microbial communities.

### 3.2. Variations in Microbial Community Composition, Diversity, and Structure

The major bacterial taxa (cumulative relative abundance > 80%) across all the AL soils belonged to the Proteobacteria (25% in average), Acidobacteria (17%), Actinobacteria (13%), Planctomycetes (12%), Gemmatimonadetes (10%), and Chloroflexi (7%) (Figure 2). By contrast, in the RL soils the Actinobacteria were detected more with mean relative abundance of 25%, followed by the Proteobacteria (21%), Acidobacteria (20%), Planctomycetes (10%), Chloroflexi (7%) and Gemmatimonadetes (6%). Among these bacterial phyla, significant differences between the AL and the RL soils were only observed in Proteobacteria and Actinobacteria (Appendix A).

Fungal phyla Ascomycota (averaged 69%), Basidiomycota (averaged 8%), Zygomycota (averaged 0.3%), Glomeromycota (averaged 0.3%), and Rozellomycota (averaged 0.1%) (Figure 2), were detected with no apparent significant differences between the two land-use types (Appendix A), except for Chytridiomycota which was significantly higher in the AL (1.4%) than in the RL (0.04%) soils (*p* = 0.009). Although soil bacterial and fungal diversity indices (Chao1, Shannon, and Simpson) did not differ significantly between the two land types (*p* > 0.05; Appendix A), the microbial community structure exhibited significant differences as represented by the distinctly separate (stress < 0.1) ellipse in the NMDS plot (Figure 3). No significant dispersion effect existed either for bacterial (*permdisp* test, *p* = 0.087) or fungal (*permdisp* test, *p* = 0.076; Figure 3) communities. The bacterial community structure differed significantly between AL and RL soils (*anosim* test, *R* = 0.618, *p* = 0.006), as did fungal community structure (*anosim* test, *R* = 0.721, *p* = 0.005; Figure 3). These variations in microbial community structures were mainly explained by variation in soil N and P, respectively (Appendix A).

### 3.3. Distinct Microbial Networks and Putative Metabolic Profiles

Prior to network comparisons, the changes in the microbial OTUs that were used for network constructions were verified. The OTU richness was over six times higher for bacterial communities (averaging 2900 OTUs) than for fungal communities (averaging 470 OTUs). The richness (Chao1) and diversity index values (Shannon and Simpson; Appendix A), indicated a higher occurrence of bacterial communities in the AL and RL soils studied. The four ecological networks were constructed using the bacterial and fungal OTUs’ correlation matrices from the AL and the RL soils (Figure 4). The similarity thresholds of the networks ranged from 0.852 to 0.940. The network topological properties such as average geodesic distance (GD), average OTU clustering coefficients (avg CC) and modularity were all higher in the ecological network than in the random network (Appendix A).

The bacterial network was more complex in the RL than in the AL soil. This is indicated by the higher numbers of OTUs (1695 in the RL vs. 1534 in the AL) and especially in the links (paired OTUs that show significant relationships) within the network (4268 in RL vs. 1888 in AL; Figure 4A). In the RL, the complex bacterial network was characterized by more cooperative bacterial taxa (as indicated by 77% positive links) than that in the AL soils. In contrast, the soil fungal network was more complex and showed a higher number of OTUs (308), links (614), and positively connected links (66%) in the AL than in the RL soils (191 OTUs, 323 links with 54% positive ones, respectively: Figure 4). Based on the higher connectivity and clustering coefficient (Table 1), the soil bacterial and fungal community structures tend to be more susceptible to potential disturbances in the RL than in the AL soils.

Based on microbial network changes, we then used PICRUSt analysis to obtain supplemental insights on potential ecosystem functionality of the sites. The relative abundance of the two strongest putative functions (amino acid and carbohydrate metabolisms) was significantly higher in RL than in AL soils (independent sample *t*-tests, *p* < 0.05, Figure 5). Additionally, the relative abundance of other putative metabolic processes involving lipids, glycan biosynthesis, xenobiotic biodegradation, terpenoids and polyketides, and amino acids, was also significantly higher in the RL than in the AL soils (independent sample *t*-tests, *p* < 0.05, Figure 4). There was a significant correlation with bacterial and to a lesser extent with fungal microbial networks and soil physicochemical properties (Table 2). The RL bacterial network was significantly related with soil C (organic C, dissolved organic C, and microbial biomass C) and N (ammonia N and total N), while the RL fungal network showed weak links with soil nutrients (Table 2). Comparatively, in AL soils, easily available N (ammonia N, nitrate N) and P (available P) significantly influenced bacterial networks, while soil salt content was the main parameter influencing AL fungal networks (Table 2).

### 3.4. Keystone Taxa and Their Distributing Feature

Based on within- and among-module connectivity of individual OTUs, an overall 115 keystone taxa were detected, with much of them assigned to bacteria (98). Specifically, the bacterial and fungal keystone taxa were both higher in the RL soils (53 and 13) than in the AL soils (45 and 4; Figure 6A).

These keystone taxa were from different phyla (Table 3). In RL soil, major keystone taxa were assigned to Proteobacteria (18), followed by Actinobacteria (12) and Acidobacteria (11), while in the AL soil, they belonged to Acidobacteria (13) and Proteobacteria (9) (Figure 6B). In RL soils, among the 53 bacterial keystone taxa (Appendix A), 9 OTUs were identified to genus level and consisted of Alphaproteobacteria (5), Planctomycetia (2), and Rubrobacteria (2); while among the 13 fungal keystone taxa, 8 were assigned at genus level, of which 1 belonged to Zygomycota and 12 to Ascomycota (Appendix A), respectively. In the AL soils, there were 4 fungal keystone taxa that belonged to Ascomycota (2), Zygomycota (1), and one unidentified fungus. There were three fungal keystone taxa belonging to the orders of Pleosporales, Hypocreales, and Saccharomycetes. Among them, the OTU 6627, OTU 2069, and OTU 3133 were assigned to parasitic or plant pathogenic fungi causing canker formation. The other three fungal keystones seem to have wide distributions. For instance, the OTU 9390 and OTU 9744 belong to the order Eurotiales which are widespread and abundant saprobic fungi [69], and the OTU 12,611 belongs to the Lecanoromycetes order (species of which seem to be common and frequently reported worldwide) [70].

Among the 45 bacterial keystone taxa (Appendix A), there were 8 genera mainly related to Alpha-, Gammaproteobacteria (5), Actinobacteria (2), and Saprospirae (1) (Table 3). The complex soil bacterial network (Figure 4) in the RL featured a condensed functional module (i.e., fewer numbers of modules, but more keystone taxa in each module). In contrast, functional modules of AL soils seemed to be more loose and composed of a higher number of modules but with single/double keystone taxa in each module (Figure 6C,D).

## 4. Discussion

### 4.1. Microbial Diversity and Community Structure Variation

At the continental scale, Lauber et al., (2009) found that soil pH was a strong predictor of soil bacterial community structure [71]. In a recent meta-analysis of impacts of global change factors on soil microbial diversity, the predominant influencing factor in land conversion (from native ecosystem to secondary ecosystem, plantation, agricultural land, and pasture) was found to be differential soil pH, rather than factorial changes associated with climate, biomes, soil resource content, or stoichiometry [72]. This partially supports our first hypothesis (H1). However, no significant differences between bacterial and fungal alpha diversity indices in soils from agricultural (AL) and rehabilitated lands (RL) were observed (Appendix A). A major reason for this deviation could be that the soil pH of the AL and the RL was not significantly different. However, the single factor of soil pH in shaping microbial communities in these soils may be confounded by the reduced effect of other soil properties, especially the SOC, TN, and MBC contents which also differed marginally in these soils [50].

In support of hypothesis 1, microbial community composition differed significantly between AL and RL soils. This was in line with the global analysis study [72], showing that land use change has a significant positive effect on microbial beta-diversity indices. In particular, the relative abundance of Proteobacteria and Actinobacteria were significantly different across the AL and the RL soils (Actinobacteria: 25% in RL vs. 13% in AL; Proteobacteria 21% in RL vs. 25% in AL). Such significant differences tend to be important ecological indicators when considering the persistence of bacterial members belonging to a specific phylum level. In the AL soils, the highest proportion of Proteobacteria (25%) might be related with their copiotrophic lifestyle [18,19] in presence of high amounts of SOC and TN in soils (Appendix A). The majority of Actinobacteria are important saprophytes and capable of breaking down a wide range of plant debris [73]. Some genera within the Actinobacteria, such as *Streptomyces* and *Micromonospora*, are renowned for their prolific production of a diverse range of bioactive metabolites including enzymes and signaling molecules used for communication within microbial communities [73,74]. Noteworthy is almost twice the relative abundance of *Streptomyces* and *Micromonospora* in the RL compared to the AL soils (*Streptomyces*: 0.2% in RL vs. 0.1% in AL; *Micromonospora:* 0.05% in RL vs. 0.03% in AL). This supports positive feedback towards bacterial physiological activity especially in the RL soils [38,75].

At the fungal phylum level, there was a significant increase in Chytridiomycota in the AL soils. This might be attributed to specific agricultural management practices such as stubble retention or reduced tillage in cultivated Loess fields [76]. This is in agreement with previous studies [21,22] describing many members of the Chytridiomycota as microscopic saprophytes capable of decomposing cellulose and chitin. The observed higher ratio of Ascomycota to Basidiomycota in AL (17) than in RL (5) (indicating a possible stronger decomposition mechanism in AL soils) is in line with a previous study linking retarded decomposition with lower Asco-/Basidomycota ratios in the alkaline soils [77] with similar pH as the agricultural soils in this study. Besides pH changes, higher water storage capacity and water content in AL soils due to reduced/no-tillage and stubble retention were reported in previously conducted studies [20,47]. The increased water content might also be due to the potential hydrophylic nature of the residual stubble and may have assisted chytrids’ abundance following their aquatic and motile lifestyle [21,22]. Meanwhile, the remarkable increase in the relative abundance of Sordariomycetes in AL soils might be attributed to wide-spread organic fertilizer use and high-efficiency compound fertilization application practices in the AL soils that are known to promote these fungi [47,78,79].

### 4.2. Distinct Microbial Networks, Keystone Taxa and Putative Metabolism Profiles

In accordance with the second hypothesis (H2), in the RL soils, bacterial taxa formed a tighter, more co-operative, and more complex network (Figure 4). Evidence has shown that complex networks are more robust than simple networks to negate the effects of environmental perturbations [80,81]. Therefore, we argue that the microbial communities in the RL soil ecosystems, with more complex networks, are better equipped with stronger resilience mechanisms against environmental stresses compared to those in the AL ecosystems. This is further supported by the observation that complex network co-ordination within different taxa can compensate for the loss of specialist taxa capable of performing particular functions in the soil matrix [82]. In the previous study, the higher complexity of microbial networks was shown to be more resilient to environmental perturbations than simple networks consisting of lower connectivity [78]. For instance, a comparative study of root microbiota in different farming systems found that the microbial network of organic farming was more complex, featuring more highly connected nodes (microbial OTUs) whereas conventional farming networks were predominated by weakly connected peripheral nodes [79].

Bacterial network complexity may be attributed to differences in the“nutrient-accumulating” patterns of soil. Typically, elevated nutrient contents (resulting from the anthropogenic nutrients inputs in AL soils) foster the preferential growth of some microbes and results in lower selection pressure for others [83]. In contrast, naturally-accumulating nutrient patterns in RL soils could act as a selective force on the assembly of the soil microbiome and increase the likelihood of coevolution [83]. In the RL, facilitated nutrient acquisition may have occurred via complex microbial interactions that are further increased by factors such as plant fine root organization, root exudates, soil aggregate build-up, as well microbial residue accumulation, as seen previously [11,38,76,84]. Although empirical evidence for this argument could not be provided in the present study, the significantly enhanced putative amino acid and carbohydrate metabolisms identified in the RL soils strongly support the idea that RL soil microbes are more engaged to recycle organic C and N substrates for the sake of plant availability.

As the Chinese Loess Plateau is a fragile habitat, the overall responses of soil microbiomes towards predicted disturbance such as extreme weather conditions or regional climate warming would be of great importance for environmental policymakers [5,33]. Therefore, land-use specific taxa that may act as a potential biological indicator for sustainable soil use and healthy ecosystems is evaluated to study the current status of these sensitive ecosystems. Using linear discriminant analysis (LDA), we identified 9 bacterial and 151 fungal land-use specific taxa. In the AL soils, *Iamia*, *Marinactinospora*, *Pyrenochaetopsis*, *Kazachstania*, and *Podospora*; in RL soils, *Chthoniobacter*, *Cenococcum*, *Tulostoma*, and *Fabrella* were high occurring biomarkers with the highest LDA scores (Appendix A). To our knowledge, this is the first study to explore microbial networks from the perspective of a large scale and longer-term restoration field experiment on the Chinese Loess Plateau. This twenty-year old restoration project has broad implications for interventions in other major dryland areas of the world.

A possible explanation for the easily-disturbed soil bacterial networks in the RL soils may be found in the positive correlation between network connectivity and soil C, N nutrients (especially the easily available fractions such as ammonia N), dissolved organic carbon, and microbial biomass carbon cycling of the soil’s easily available nutrients is shown to be sensitive to disturbances, and this is a probable factor shaping the microbial communities and their connectivity [85,86,87]. Conversely, in the AL soils fungal networks were less disturbed, probably due to cascading interlinks within the fungal communities themselves [88,89] which tend to be relatively stabilized under changing available nutrient patterns. Furthermore, the higher network complexity (as shown by the higher number of taxa and the higher number of associations that those taxa shared among them) might also have contributed to the corresponding resistance and resilience of fungal communities towards periodic land management in the AL soils. In the soil microbiome, bacterial and fungal keystone taxa have been computationally inferred using network scores [90,91,92]. For a range of taxa, it has been shown that keystone taxa identified using statistical tools do indeed have an impact on the composition and performance of the microbiome [91,92,93,94]. Microbial keystone taxa have been widely identified in agroecosystems [61,95,96], but not in the RL ecosystems of the Loess region. Keystone taxa detected in the RL soils, such as *Rubrobacter*, *Balneimonas*, and *Planctomycetes*, play a significant role in resistance mechanisms, oxidation, and nutrient acquisition, especially in the soil N cycling [97,98]. All this is indicative of a different mechanism of microbial management and enrichment of soil conditions at the land-use level.

In comparison, the bacterial keystone taxa in the AL soils were substantially different from those detected in the RL soils (Table 3). The common occurrence of *Rhodoplanes* in both land-use type soils may suggest that they could be regarded as the indicator species for these soils. More investigation is required on which members of this bacterial genus and their habitat patterns can help predict changes in the community composition at ecosystem level. Except for *Rhodoplanes,* others belonged to various and varied bacterial genera and therefore make specific explanations regarding bacterial role in these soils very difficult. However, the genus *Flavisolibacter* has previously been reported in cultivated soils [99]. The genera *Aeromicrobium* and *Kribbella* (belonging to the Actinobacteria) have also been isolated from alkaline soils [100,101] similar to Loess alkaline agricultural soils. The specific functions of these keystone taxa still remain largely unknown.

In our studied AL soils, there were only two identified fungal keystone taxa; one was unidentified Microascaceae (OTU 13053), and the others members from family Microascaceae, which mostly consist of saprobes and plant-pathogenic genera include *Microascus* and *Pseudallescheria* [102]. Another fungal keystone was *Talaromyces marneffei* (OTU 5193) from the genus *Talaromyces*. *T. marneffei* is the known dimorphic species producing filamentous growth and yeast phase at different temperatures, indicating its strong reproductive strategy. Moreover, *T. marneffei* is an emerging fungal pathogen causing mycosis in an immune-compromised East Asian population [103] whose occurrence in the Loess barren agricultural soils is an early indication of probable health risk factor from these soils. On the other hand, under organic farming practices, Banerjee, Walder, et al., (2018) reported majority of key-stone taxa belonging to arbuscular mycorrhizal fungi (AMF) and further to fungal orders that can form neutral and beneficial interactions with plants.

Fungal keystone taxa *Mortierella* (belonging to the order of Mortierellales) are the module connectors (with among-module connectivity (Pi) > 0.62) present in both the AL and the RL soils. Fungal genera *Mortierella* are characterized by (i) high ecological and physiological diversity enabling them to be distributed worldwide [104]; and (ii) lipid production that can be potentially used for bacterial biomass incorporation [105]; while *Phaeosphaeria*, *Penicillium*, *Phaeophyscis*, *Geopora*, *Kazachstania*, *Acremonium*, and *Fusarium* are unique keystone genera in RL soils and the identified fungal keystones were assigned to saprotrophs and symbiotrophs [106]. Specifically, the two fungal keystone taxa (OUT 56 and OTU 4181, that belong to the order Pezizales) are mostly saprophytic in soil rich in humus and other plant residues, indicating their roles in soil nutrient accumulation during soil restoration. Altogether, this strongly supports our second hypothesis (H2) expecting more keystone taxa embedded in a more complex microbial network in rehabilitated land (RL) than in the agricultural land (AL). How these keystone taxa interact within the microbial network, and how that changes the overall community function over time, remains the subject of future studies.

## 5. Conclusions

Our study revealed significant changes in microbial community composition, especially that of the bacterial members, in the agricultural and rehabilitated soils of the Chinese Loess Plateau. A more complex yet co-operative bacterial network was observed in the rehabilitated soils. Regular land management practices such as fertilizer application may also have reduced the microbial need for nutrient cycling in the agricultural soils, thus resulting in a less diverse bacterial network. Moreover, in agricultural ecosystems, soils are more prone to the artificial management practices leading to increased disturbance in the soil micro-environment. To cope with such frequent disturbances, the fungal communities needed to be more diverse and cooperative, as supported by higher diversity index and fungal network attributes. Investigating changes in such keystone taxa can provide the basis for future experiments studying how microbe-informed soil management practices may lead to the effective restoration and improvement methodologies of eroded lands, particularly in dryland systems such as those in the Loess plateau region.

## Figures and Tables

**Figure 1 biology-10-01261-f001:**
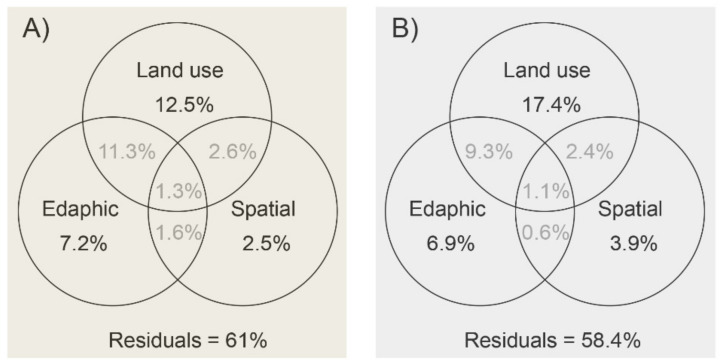
Venn diagram representing variation partitioning of bacterial (**A**) and fungal (**B**) communities explained by land-use types, and edaphic and spatial variables.

**Figure 2 biology-10-01261-f002:**
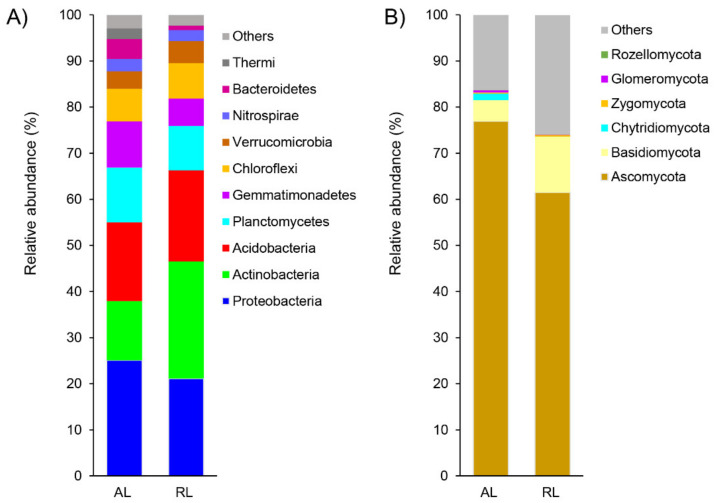
Relative abundance of the main bacterial (**A**) and fungal (**B**) phyla in agricultural (AL) and rehabilitated lands (RL) of the Chinese Loess Plateau.

**Figure 3 biology-10-01261-f003:**
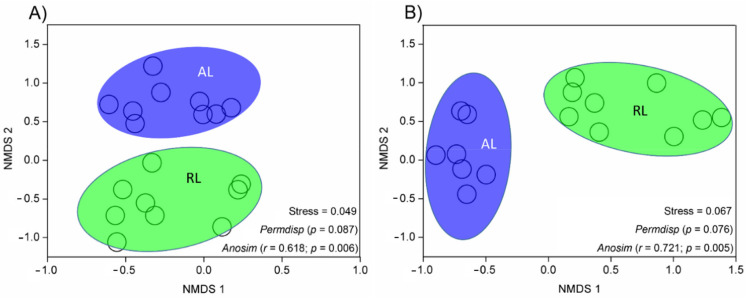
Soil bacterial (**A**) and fungal (**B**) community structures indicated by unweighted Non-Metric Multi-Dimensional Scaling (NMDS) plots of pairwise UniFrac community distance across agricultural (AL) and rehabilitated lands (RL) of the Chinese Loess Plateau. The stress value determines the consistency of the new model (ranks on the ordination configuration) with the original data (ranks in the original similarity matrix); a model with stress < 0.1 was accepted. Permutation test for homogeneity of multivariate dispersion (Permdisp) was used to test heteroscedasticity between groups. Non-parametric Anosim (analysis of similarities) based on community distance matrix indicates significant differences between the grouping factor (land-use type) and the degree (*r*) of microbial community composition. The higher the r value, the higher is the interpretation of the difference.

**Figure 4 biology-10-01261-f004:**
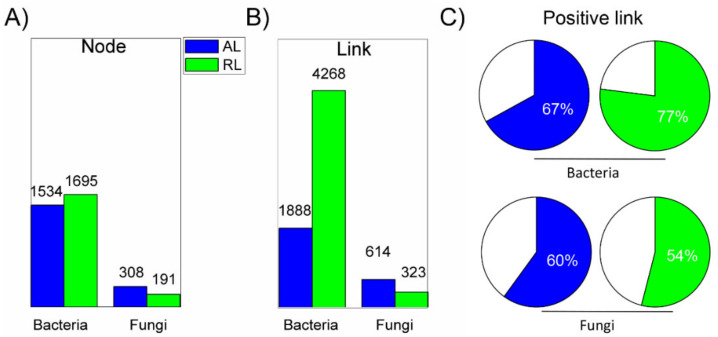
Potential interactions in microbial networks in agricultural (AL) and rehabilitated lands (RL) of the Chinese Loess Plateau. Networks were constructed based on a random correlation matrix of the OTUs. Complexity and interaction of bacterial and fungal OTUs within the microbial networks, indicated by (**A**) number of nodes (**A**), significant associations/links among the OTUs (**B**) and positive relationship/links among the detected OTUs (**C**).

**Figure 5 biology-10-01261-f005:**
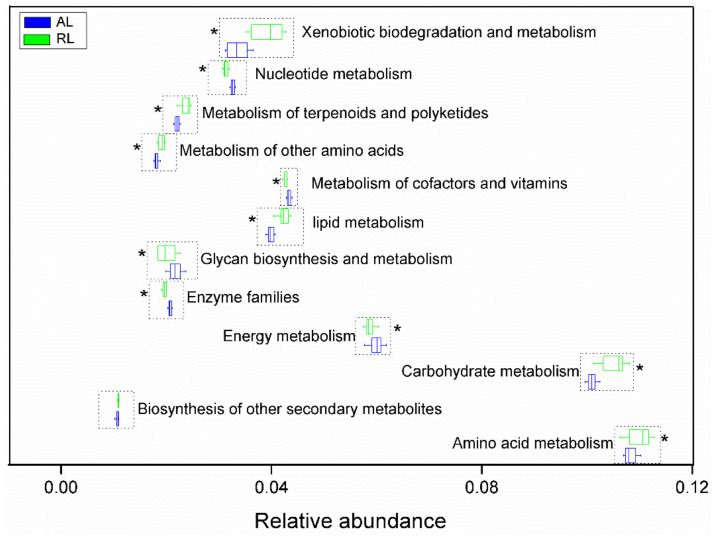
Metabolic profiling of bacterial communities, based on PICRUSt 2 analysis, in the agricultural (AL) and rehabilitated lands (RL) of the Chinese Loess Plateau. Asterisks indicate significant differences in the predicted metabolic function of the OUT-based bacterial community analysis at *p* < 0.05 significance level in independent sample *t*-tests.

**Figure 6 biology-10-01261-f006:**
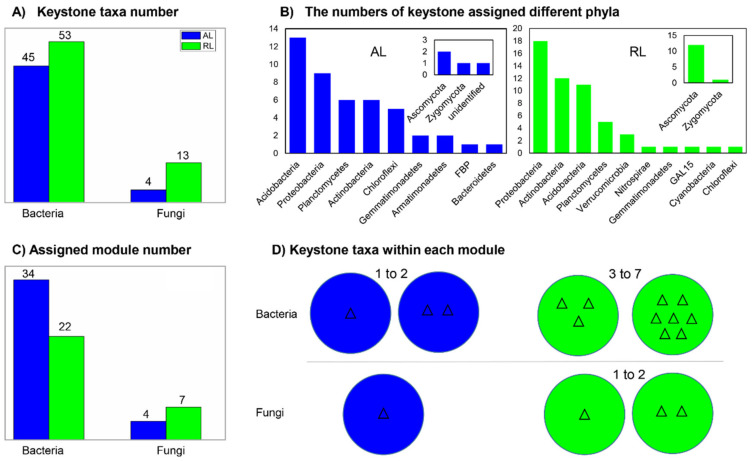
Keystone microbial taxa network and distribution characteristics in agricultural land (AL) and rehabilitated land (RL) soils of the Chinese Loess Plateau. The number of keystone taxa (**A**), assigned phyla to the number of keystone taxa (**B**), assigned functional module to the number of keystone taxa (**C**) and, distribution features of the number of keystone taxa (**D**) based on random correlation matrix analysis of the OTUs.

**Table 1 biology-10-01261-t001:** Topological parameters of the empirical molecular ecological networks (MENs) of bacterial and fungal communities in agricultural (AL) and rehabilitated lands (RL) of the Chinese Loess Plateau.

	Bacteria		Fungi	
	AL	RL	AL	RL
Threshold	0.96	0.96	0.89	0.87
avgCC ^a^	0.08	0.10	0.12	0.16
avgK	2.46	5.04	3.39	3.98
Module	246	204	39	25
Modularity	0.94	0.77	0.68	0.63
GD	9.58	6.33	5.14	4.35

^a^ Abbreviations: avgCC = average clustering coefficient of nodes; avgK = average connectivity among nodes; GD = average geodesic distance between nodes.

**Table 2 biology-10-01261-t002:** Mantel test results indicating correlations between soil physicochemical properties, and bacterial and fungal molecular ecological networks in agricultural (AL) and rehabilitated lands (RL) of the Chinese Loess Plateau.

	Bacteria				Fungi			
	AL		RL		AL		RL	
	*r*	*p*	*r*	*p*	*r*	*p*	*r*	*p*
pH	0.010	0.269	0.015	0.125	0.017	0.278	0.071	0.121
^a^ TOC	0.005	0.382	0.034	**0.039**	0.026	0.181	0.068	0.114
TN	0.006	0.371	0.082	**0.001**	0.005	0.388	0.080	0.120
TP	−0.001	0.494	0.003	0.385	−0.038	0.881	0.112	0.052
Moisture	−0.015	0.775	−0.016	0.831	−0.011	0.610	0.047	0.146
SC	0.009	0.273	0.033	0.997	0.076	**0.018**	0.000	0.370
NO_3_	0.103	**0.001**	−0.033	0.994	−0.016	0.631	−0.049	0.903
NH_4_	0.041	**0.023**	0.068	**0.001**	−0.045	0.897	0.035	0.232
AP	0.052	**0.004**	0.047	0.005	0.023	0.230	0.017	0.264
DOC	0.006	0.301	0.007	**0.002**	0.005	0.381	0.052	0.142
MBC	−0.027	0.937	0.081	**0.001**	−0.035	0.850	0.021	0.241
Altitude	0.325	0.211	0.314	0.240	0.341	0.193	0.329	0.224
Plant type	0.355	0.153	0.345	0.194	0.369	0.124	0.355	0.172

^a^ Abbreviations: AP, available phosphorus; DOC, dissolved organic carbon; MBC, microbial biomass carbon; SC, salt content; TOC, total organic carbon; TN, total nitrogen; TP, total phosphorus. Bold numbers show those with *p* values < 0.05.

**Table 3 biology-10-01261-t003:** Keystone bacterial and fungal taxa detected in agricultural lands (AL) and rehabilitated lands (RL) on the Chinese Loess Plateau. Keystone taxa in Table 3 is the subset of Appendix A but with most possible taxonomic assignment for the identified keystone taxa, i.e., up to the genus level in terms of bacterial and species level in fungal taxa, and that the full list of keystone taxa from the analyses can be found in Appendix A.

	OTUs	Kingdom	Class	Order	Family	Genus	Species
AL	OTU69876	Bacteria	Saprospirae	[Saprospirales]	Chitinophagaceae	*Flavisolibacter*	*Not assigned*
AL	OTU72155	Bacteria	Actinobacteria	Actinomycetales	Nocardioidaceae	*Aeromicrobium*	*Not assigned*
AL	OTU37190	Bacteria	Actinobacteria	Actinomycetales	Nocardioidaceae	*Kribbella*	*Not assigned*
AL	OTU5872	Bacteria	Alphaproteobacteria	Rhizobiales	Hyphomicrobiaceae	*Devosia*	*Not assigned*
AL	OTU1603	Bacteria	Alphaproteobacteria	Sphingomonadales	Sphingomonadaceae	*Kaistobacter*	*Not assigned*
AL	OTU46084	Bacteria	Alphaproteobacteria	Rhizobiales	Hyphomicrobiaceae	*Rhodoplanes*	*Not assigned*
AL	OTU53987	Bacteria	Gammaproteobacteria	Pseudomonadales	Pseudomonadaceae	*Pseudomonas*	*Not assigned*
AL	OTU36507	Bacteria	Gammaproteobacteria	Xanthomonadales	Sinobacteraceae	*Steroidobacter*	*Not assigned*
RL	OTU40252	Bacteria	Alphaproteobacteria	Rhizobiales	Bradyrhizobiaceae	*Balneimonas*	*Not assigned*
RL	OTU46298	Bacteria	Alphaproteobacteria	Rhizobiales	Brucellaceae	*Ochrobactrum*	*Not assigned*
RL	OTU65537	Bacteria	Alphaproteobacteria	Rhizobiales	Hyphomicrobiaceae	*Rhodoplanes*	*Not assigned*
RL	OTU60443	Bacteria	Alphaproteobacteria	Rhizobiales	Hyphomicrobiaceae	*Rhodoplanes*	*Not assigned*
RL	OTU6889	Bacteria	Alphaproteobacteria	Rhizobiales	Hyphomicrobiaceae	*Rhodoplanes*	*Not assigned*
RL	OTU28903	Bacteria	Planctomycetia	Pirellulales	Pirellulaceae	*A17*	*Not assigned*
RL	OTU3183	Bacteria	Planctomycetia	Planctomycetales	Planctomycetaceae	*Planctomyces*	*Not assigned*
RL	OTU6858	Bacteria	Rubrobacteria	Rubrobacterales	Rubrobacteraceae	*Rubrobacter*	*Not assigned*
RL	OTU14570	Bacteria	Rubrobacteria	Rubrobacterales	Rubrobacteraceae	*Rubrobacter*	*Not assigned*
AL	OTU5193	Fungi	Eurotiomycetes	Eurotiales	Trichocomaceae	*Talaromyces*	*T. marneffei*
AL	OTU13116	Fungi	*Incertae sedis*	Mortierellales	Mortierellaceae	*Mortierella*	*M. indohii*
AL	OTU13053	Fungi	Sordariomycetes	Microascales	Microascaceae	*unidentified*	*unidentified*
AL	OTU11811	Fungi	unidentified	unidentified	unidentified	*unidentified*	*unidentified*
RL	OTU6627	Fungi	Dothideomycetes	Pleosporales	Phaeosphaeriaceae	*Phaeosphaeria*	*unidentified*
RL	OTU10400	Fungi	Dothideomycetes	Pleosporales	unidentified	*unidentified*	*unidentified*
RL	OTU9390	Fungi	Eurotiomycetes	Eurotiales	Trichocomaceae	*Penicillium*	*P. polonicum*
RL	OTU9744	Fungi	Eurotiomycetes	unidentified	unidentified	*unidentified*	*unidentified.*
RL	OTU7322	Fungi	*Incertae sedis*	Mortierellales	Mortierellaceae	*Mortierella*	*M. humilis*
RL	OTU12611	Fungi	Lecanoromycetes	Teloschistales	Physciaceae	*Phaeophyscia*	*P. hispidula*
RL	OTU4181	Fungi	Pezizomycetes	Pezizales	Pyronemataceae	*Geopora*	*unidentified*
RL	OTU56	Fungi	Pezizomycetes	Pezizales	Pyronemataceae	*unidentified*	*unidentified*
RL	OTU6753	Fungi	Saccharomycetes	Saccharomycetales	Saccharomycetaceae	*Kazachstania*	*K. telluris*
RL	OTU2069	Fungi	Sordariomycetes	Hypocreales	*Incertae sedis*	*Acremonium*	*A. dichromosporum*
RL	OTU3133	Fungi	Sordariomycetes	Hypocreales	Nectriaceae	*Fusarium*	*F. pseudensiforme*
RL	OTU6835	Fungi	unidentified	unidentified	unidentified	*unidentified*	*unidentified*
RL	OTU6835	Fungi	unidentified	unidentified	unidentified	*unidentified*	*unidentified*

OTU = Operational Taxonomical Unit.

## Data Availability

Not applicable.

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
