# Peer review of "Soil Rehabilitation Promotes Resilient Microbiome with Enriched Keystone Taxa than Agricultural Infestation in Barren Soils on the Loess Plateau"

_biology, 2021, doi:10.3390/biology10121261_

Round 1
Reviewer 1 Report
The study focused on investigating the changes of these key microbes in dryland barren soils compare with those in agricultural soils. Liu and his collagues found that plant restoration favors the establishment of a more complex soil microbial network that provides positive feedback to nutrient accumulation Authors have done an interesting and integrated work, and I enjoy reading this manuscript. Overall, the experiments are well designed and a variety of statistical analysis methods are sufficient and appropriate. The Figures are eye-catching as well. Therefore, I only have some comments to improve the manuscript. For the fig.2, I would suggest to give the results of all soil samples rather than their pooled Fig S1 is not necessary for the study. I suggest to further improve the English of the ms. Fig.6, pls remove the arrows.Author Response
Answers to Reviewers comments
Comment 1
For the fig.2, I would suggest to give the results of all soil samples rather than their pooled.
Answer: The relative abundances of the bacterial and the fungal phyla in Fig 2 were calculated using all soil samples. For individual samples i.e. not pooled, the values are presented in Table S2 and Table S3.
Comment 2
Fig S1 is not necessary for the study.
Answer: As per the suggestions, the previous Figure S1 has been removed, instead, we include a new Figure S1 showing bacterial and fungal diversity indices in accordance with the suggestions of reviewer 2.
Comment 3
I suggest to further improve the English of the ms.
Answer: The manuscript text is proof-read for the improvement in the English language by native English speaker.
Comment 4
Fig.6, pls remove the arrows.
Answer: Following the suggestions, the arrows in Figure 6 are removed.
Reviewer 2 Report
Reviewer’s comments
In this manuscript, the authors conducted a metagenomic study to investigate soil microbial profiles between agriculture land (AL) and rehabilitated land (RL) in the Loess Plateau of China. The authors showed distinct soil microbial profiles affected by the land use type (AL versus RL). Then, the authors performed molecular microbial network analyses between AL and RL. They found more significant relationships among bacterial taxa in RL than AL, but more significant relationships among fungal taxa in AL than RL. In addition, the analyses indicated that the microbial network in RL has more disturbance and resilience, with more metabolic profiles, than the network in AL. The authors also found significant correlations between soil nutrient conditions and the bacterial networks. Finally, the authors found higher keystone taxa in RL compared to AL. This study serves as a foundation to understand microbial networks in these barren soils, which can lead to further development of land use in the area.
Overall, I think the storyline is well elaborated. Most analyses are suitable to address the authors’ questions. However, there are a few points the authors need to be careful in data interpretation and conclusion. Please see below for my comments:
Major comments:
- I strongly recommend the authors to reconsider the title of the manuscript. When I first read the title “Plant Restoration Enhances Microbial Keystone Taxa in Barren Soils on The Loess Plateau”, I think the authors should compare microbial profiles between unrestored barren soils and restored soils to demonstrate that introduced vegetation enhances microbial keystone taxa. However, I do not see this part in the manuscript as the authors compared only between AL and RL microbial community. In addition, the word ‘plant restoration’ seems ambiguous to me as both agriculture and rehabilitation involve introducing plants to barren soils. Finally, the authors have several findings in their study, but the title does not have a good coverage as it only mentions about keystone taxa. I would suggest something like “Soil Rehabilitation promotes resilient microbiome with enriched keystone taxa than agricultural infestation in barren soils on the Loess plateau”.
- Have the authors tested other confounding factors to verify that they do not account for microbial profiles? For instance, the authors describe that a half of AL studied sites are in low altitudes while the rest of studied sites are in high altitudes. Would the altitude influence the microbial profiles among AL sites? Furthermore, there is a variety of vegetation across studied sites. Could this account for different microbial profiles? Is there any difference due to types of plants (tree versus herbaceous species)? These tests would be a good support to ensure that the authors’ findings are primarily due to different land use type (AL versus RL). It would be also great if the authors have data from native barren soil to compare, but I understand that there is a limitation to retrieve the data.
- In reality, any microbes can interact with each other no matter where they come from. Thus, bacteria can also interact with fungi. I am not sure why the authors conduct separated analyses for bacteria and fungi, instead of constructing the microbial networks for the whole microbiome. It would be great if the authors can provide a justification in the manuscript.
- I feel a disconnection between the proposed hypotheses in the introduction and in the discussion. For example, in line 369ff, I wonder if pH is a part of ‘soil nutrient pools’ as the authors mention in H1 at the introduction. For H2, the introduction states about taxa connectivity, network connectivity and metabolic profiles. However, the discussion (line 377ff) says about relative abundance and beta diversity. Finally, H3 in the introduction states about the number of keystone taxa, assigned phyla and functional modules between AL and RL. However, in the discussion (line 472ff) the authors do not compare those indices with the data generated from this study, but instead talk about the prevalence patterns of fungal keystone taxa in different land types to support the proposed H3. Overall, the authors should have concurrent hypotheses in the introduction and the discussion to avoid readers’ confusion.
Minor comments:
- It would be interesting if the authors could layout any shared/unique keystone taxa between two land types (AL versus RL). If there are common taxa, representing data as a Venn diagram may be useful to highlight complexity/uniqueness of microbial networks in each land use type.
- Line 46: Should the authors clarify that this is for human population? Or do the authors mean the population of any organisms?
- Line 58: Please spell out ‘hectares’, otherwise please specify an abbreviation.
- Line 65-67: In these two references, did they truly conduct experiments to test that these microbes secrete enzymes that contribute to nutrient accumulation? Didn’t they just perform association analyses between presence of microbes and nutrient conditions? If the latter, I do not think these two references provide a strong support for the statement.
- Line 67: Abundance of what? Enzyme, nutrient condition, or microbe?
- Line 68ff: Why Chytridiomycota? Other groups of fungi can also efficiently decompose cellulose and chitin such as Agaricomycotina or Pezizomycotina. I am not sure if references 22-24 really support the statement as those are not experimental studies showing the case.
- Line 73 – 74: I am not sure if I understand the sentence correctly. Do the authors mean that as keystone taxa affect many microbial associations so they primarily contribute to changes in soil microenvironment?
- Line 79 – 81: Please provide a citation to support this statement. I would recommend rewriting the sentence to something like “Several studies showed that changes in land use affect changes in soil microbial community and their respective keystone taxa.”
- Line 82: Should this be “keystone microbes”? Or rewrite to something like “keystone microbes are significantly increased in vegetation-restored soils …”.
- Line 104ff: The authors claim that this microbiome study reflects changes in a “longer-term restoration field experiment”. When I first read this, I expect to see the authors collecting soil data in a time-series fashion to track changes overtime between RL and AL land type. However, I just saw the data collection from one time point. I would recommend the authors to rewrite this sentence to avoid confusing the readers.
- Line 130: I would recommend saying “stored at -20 C prior PCR amplification according to Liu et al. (2018b).”
- Line 140ff: Please clarify whether processing of sequencing reads (like adapter and low-quality base trimming) is required before subsequent analyses.
- Line 141 – 143: How did the authors normalize the reads? For example, did the authors just randomly pick 49,729 reads in the sample that has larger sequencing reads? Did the authors check whether this trimming process affects the microbial composition?
- Line 144: Did the authors have a criterion for ‘high quality sequences’? Please specify.
- Line 200: Please provide a citation for the vegan package.
- Line 275: I do not see Figure S1 showing Chao1 and Shannon Simpson diversity in the manuscript package. Please provide it a supplementary material.
- Line 326ff: It would be great if the authors provide a Zi-Pi plot (as a supplementary figure) to illustrate the occurrence of keystone taxa.
- Line 374: Should it be ‘confounded’?
- Line 369ff: This is hard to judge as the authors do not provide the alpha diversity data (Figure S1) in this manuscript. Probably the authors can help the readers visualize the data by making a plot showing that there is no relationship between soil pH and alpha diversity for each site.
- Line 397 – 402: Again, Ascomycota and Basidiomycota are effective cellulose and lignin degraders. I am not sure that this explains why chytrids are more abundant in AL soil. RL soils also have cellulose biomass for microbiome community. Would it be possible that AL soils have more soil moisture content (due to watering) than RL soils so that AL soils attract higher chytrid abundance? Chytrids are often referred as ‘water mold’ so they prefer high moisture for their growth. The authors should have checked the moisture content between AL and RL soil.
- Line 405ff: I am not sure if this statement is strong unless the authors have the data showing that Sordariomycetes taxa existing in the AL soils represent a diverse array of nutritional modes (plant pathogens, saprobes, epiphytes, coprophiles and fungicoles). Seeing a higher abundance of only ‘Sordariomycetes’, but not at genus or species levels, does not mean that it represents a diverse functionality in the ecosystem.
- Line 416ff: Could the authors demonstrate this with their existing data? That would make a stronger support.
- Line 434 – 435: Could the authors make a summarized table showing this ‘land-use specific taxa’? That would be very useful for the readers.
- Line 437: I think the authors should provide more details on study sites elsewhere in the manuscript (maybe in Materials and Methods). For example, how long have those AL, RL sties been existed? How have they been treated? Providing this info would explain the ‘long term restoration field’ aspect of the current study.
- Line 463 – 464: Is it possible that the finding shows an indicator species for this locality, this climate type or something else?
- Line 473: Please define how the authors assign Mortierella as generalists based on their data. Please also italicize the genus name Mortierella.
- Line 480ff: I am skeptical how the authors make a conclusion here. The authors mention only keystone taxa in RL, but not in AL. Then, the authors cite other studies that found AMF in AL and conclude that keystone taxa in RL are different from AL. I think the authors should have used data from their study for comparison and inference.
- Line 494 – 495: The avgK values from Table 1 show that RL soils have more disturbance than AL soils, which seems counterintuitive with this statement.
- Table 3 and S6: It looks like Table 3 presents a subset of keystone taxa from Table S6. Please clarify the difference between Table 3 and Table S6. Probably the authors subset only taxa with taxonomic assignment up to the family level for Table 3?
- Table S1: Please provide the data for soil moisture content in the table.
- Table S4: NMDS is non-metric score. I think it would be more suitable if the authors use Spearmann’s rank correlation as a non-parametric test, instead of Pearson’s correlation.
- Table S6: Please indicate that ‘B-‘ stands for bacterial taxa and ‘F-‘ stands for fungal taxa.

Author Response
In this manuscript, the authors conducted a metagenomic study to investigate soil microbial profiles between agriculture land (AL) and rehabilitated land (RL) in the Loess Plateau of China. The authors showed distinct soil microbial profiles affected by the land use type (AL versus RL). Then, the authors performed molecular microbial network analyses between AL and RL. They found more significant relationships among bacterial taxa in RL than AL, but more significant relationships among fungal taxa in AL than RL. In addition, the analyses indicated that the microbial network in RL has more disturbance and resilience, with more metabolic profiles, than the network in AL. The authors also found significant correlations between soil nutrient conditions and the bacterial networks. Finally, the authors found higher keystone taxa in RL compared to AL. This study serves as a foundation to understand microbial networks in these barren soils, which can lead to further development of land use in the area.
Answer:We are extremely thankful to the reviewer for their constructive comments and high degree of recognition towards the significance of our research work.
Overall, I think the storyline is well elaborated. Most analyses are suitable to address the authors’ questions. However, there are a few points the authors need to be careful in data interpretation and conclusion. Please see below for my comments:
Major comments:
- I strongly recommend the authors to reconsider the title of the manuscript. When I first read the title “Plant Restoration Enhances Microbial Keystone Taxa in Barren Soils on The Loess Plateau”, I think the authors should compare microbial profiles between unrestored barren soils and restored soils to demonstrate that introduced vegetation enhances microbial keystone taxa. However, I do not see this part in the manuscript as the authors compared only between AL and RL microbial community. In addition, the word ‘plant restoration’ seems ambiguous to me as both agriculture and rehabilitation involve introducing plants to barren soils. Finally, the authors have several findings in their study, but the title does not have a good coverage as it only mentions about keystone taxa. I would suggest something like “Soil Rehabilitation promotes resilient microbiome with enriched keystone taxa than agricultural infestation in barren soils on the Loess plateau”.
Answer: We agree with the reviewer’s identification of “plant restoration” to be a bit ambiguous. Therefore, we accept and appreciate the efforts for suggesting a more appropriate title for our work “Soil rehabilitation promotes resilient microbiome with enriched keystone taxa than agricultural infestation in barren soils on the Loess plateau” which replaces the earlier title,” Plant Restoration Enhances Microbial Keystone Taxa in Barren Soils on The Loess Plateau”.
- Have the authors tested other confounding factors to verify that they do not account for microbial profiles? For instance, the authors describe that a half of AL studied sites are in low altitudes while the rest of studied sites are in high altitudes. Would the altitude influence the microbial profiles among AL sites?
Furthermore, there is a variety of vegetation across studied sites. Could this account for different microbial profiles? Is there any difference due to types of plants (tree versus herbaceous species)? These tests would be a good support to ensure that the authors’ findings are primarily due to different land use type (AL versus RL). It would be also great if the authors have data from native barren soil to compare, but I understand that there is a limitation to retrieve the data.
Answer:
The reviewer’s valuable comments in these regards are much appreciated. The confounding factor was tested by Variation Partitioning Analysis (VPA) (Fig. 1). Spatial variables solely explained only 2.5% and 3.9% of variations for soil bacterial and fungal communities, respectively. The VPA indicated that the total microbial community variation was mainly explained by the factor land-use type (Bacteria, 27.7% and Fungi, 30.2%, respectively).
Within each land-use type (i.e. AL or RL), we employed Mantel’s test to evaluate the effect of confounding factors (i.e. altitude or plant type) on soil bacterial and fungal communities structures. The results showed that among the studied sites, altitude variation did not significantly affect soil bacterial (r = 0.325, p = 0.21 for AL; r = 0.314, p = 0.24 for RL) and/or fungal communities (r = 0.341, p = 0.19 for AL; r = 0.329, p = 0.22 for RL). Within the investigated sites, influence from types of plants (i.e. trees or herbaceous species) was also not obvious for microbial communities (Bacteria, r = 0.355, p = 0.15; fungi, r = 0.369, p = 0.12 for RL; Bacteria, r = 0.345, p = 0.19; fungi, r = 0.355, p = 0.17 for RL).
These results are coherent, since in AL sites, soils were subjected to long-term intensive agricultural use and exhibited weaker influence from their distributed altitudes. Among RL sites, a minor influence from plant type might also be attributed to the fact that we selected typical vegetative types’ of soils particularly with similarities in terms of; i) similar parent material of the Loessial soil group (IUSS, 2014), and ii) similar land rehabilitation starting point from the initiation of Grain for Green (GFG) project (year).
Unfortunately, regarding the native barren soils, we regret that during our soil sampling period, the majority of land area was artificially transformed, which limited us from having information about pure native barren soils even if the intent was only for the statistical use of such data. Nevertheless, we admit that such comparisons would be more meaningful and solidify the final conclusions and should be done wherever possible.
- In reality, any microbes can interact with each other no matter where they come from. Thus, bacteria can also interact with fungi. I am not sure why the authors conduct separated analyses for bacteria and fungi, instead of constructing the microbial networks for the whole microbiome. It would be great if the authors can provide a justification in the manuscript.
Answer: We agree that soil bacteria, fungi, oomycetes and other microorganisms mutually interact within a complex network.
However, our understanding is that even though, the bacteria and fungi are central to biogeochemical processes, they may follow diverse trajectories during ecosystem development and especially in situations of ecosystem recovery from stress or disturbance. Bacteria are fast growers compared to fungi who in turn are more strongly influenced by vegetation and more tolerant to environmental changes. Moreover, although an integral part of soil microbial consortium, bacteria differ from fungi mainly in terms of their life history and phenological characteristics. Furthermore, the variations in the bacterial community structure might be evident in short term and may represent highly complex networking abilities, whereas fungi survive over longer time periods and may have simpler networking abilities than bacteria. Therefore, constructing microbial networks for the whole microbiome might give a mixed picture of the reality and thus may restrict future studies whose main interest may consist of studying the successional patterns of the individual microbial groups. Therefore, to facilitate separate information and considering the differences in these two important soil microbial communities, their network analyses were conducted separately in our presented work.
In our experience (https://doi.org/10.1128/mSphere.00039-21), if a study is focused on visualizing microbial co-occurrence pattern, a composite microbial networks for the whole microbiome would be highly recommended. Whereas, for keystone taxa-focused researches, individual microbial networks are more appropriate based on the facts that;
1) In principle, because of the sequence coverage issue, the bacterial and fungal OTU numbers may differ greatly, and this could hide/override the number of keystones within the low-numbered microbial network. For instance, we checked the changes in the microbial OTUs that were used for network constructions. The OTU richness was over 6 times higher for bacterial communities (averaging 2900 OTUs) than that for the fungal communities (averaging 470 OTUs). A pooled network was constructed prior to network comparison; however, the fungal keystones were masked and cannot be identified properly.
2) Changes in bacterial communities link more strongly to soil functioning during recovery than the changes in fungal communities. Therefore their networks were separately compared in response to the disturbances (10.1038/s41467-018-05516-7).
3) For comparable studies conducted in Chinese soils, it was suggested that distinct mechanisms shape soil bacterial and fungal networks (10.1093/femsec/fiaa030). Furthermore, the soil bacterial and fungal communities showed distinct recovery patterns during restoration (10.1128/AEM.00966-17).
Foreseeing these reasons, separate analyses for the bacteria and the fungal networks were performed in our study. Additionally, methodological limitations should also be considered which need to be discussed modestly when possible.
- I feel a disconnection between the proposed hypotheses in the introduction and in the discussion. For example, in line 369ff, I wonder if pH is a part of ‘soil nutrient pools’ as the authors mention in H1 at the introduction.
Answer: The previous hypothesis 1 (H1) “We hypothesized that there would be significant variations in microbial diversity and community structure between RL and AL soils (H1) due to the closer interrelationships between the soil nutrient pools and the microbial community [35].” has been modified to “The soil properties and microbial communities are interlinked [34,35] and therefore in our H1 hypothesis we expect significant variations in the microbial diversity and community composition between the RL and the AL land-use type soils.” in addition, we included a new reference [35] as belowï¼›
[35] Liu, T.; Wu, X.; Li, H.; Alharbi, H.; Wang, J.; Dang, P.; Chen, X.; Kuzyakov, Y.; Yan, W. Soil organic matter, nitrogen and pH driven change in bacterial community following forest conversion. For. Ecol. Manage. 2020, 477, doi:10.1016/j.foreco.2020.118473.
For H2, the introduction states about taxa connectivity, network connectivity and metabolic profiles. However, the discussion (line 377ff) says about relative abundance and beta diversity.
Answer: We apologize for the inaccurate information presented in the earlier text. this part of the discussion was related to H1 and not H2. We have rectified, “Despite this, in line with the second hypothesis (H2)” to “In support of Hypothesis 1”.
Finally, H3 in the introduction states about the number of keystone taxa, assigned phyla and functional modules between AL and RL. However, in the discussion (line 472ff) the authors do not compare those indices with the data generated from this study, but instead talk about the prevalence patterns of fungal keystone taxa in different land types to support the proposed H3. Overall, the authors should have concurrent hypotheses in the introduction and the discussion to avoid readers’ confusion.
Answer: To have concurrent hypotheses in the introduction and avoid confusion, we have added some new references and re-worked on the previous hypotheses.
“Additionally, taxa and network connectivity, depicting relationships in microbial communities, are expected to be more enhanced, with richer microbial metabolic pro-files, in rehabilitated land (H2). Finally in our third hypothesis (H3), we speculate that the number of keystone taxa, assigned phyla, and functional modules to the keystone taxa, would be higher in AL than in RL soils.” In the earlier text at line ….
has been re-written as
“Carbon through differential input land likewise the other important nutrients such as the N, P, S are accumulated differently in the diverse vegetative habitats/land-use types. In the AL land-use type, soil microbes have substrate-competitive pressure due to constantly anthropogenic nutrients inputs [36]. Whereas, in RL, without artificial disturbances, soil nutrients accumulate more naturally and there is higher C input in the RL soils as generally seen in the forest ecosystems [37, 38] and hence the microbial-mediated nutrient build-up, especially for bacterial microbiota, as soil bacteria are regarded as more important mediators for rapid nutrient cycle [39]. Moreover, during forest ecosystem recovery, bacterial community interactions were complex whereas the fungal networks were seen to be relatively simpler and more isolated [40, 41]. Based on these understandings, in our second hypothesis (H2) we expect a more complex and cooperative bacterial network in rehabilitated land (RL), along with more keystone taxa and richer functional modules.”
Minor comments:
- It would be interesting if the authors could layout any shared/unique keystone taxa between two land types (AL versus RL). If there are common taxa, representing data as a Venn diagram may be useful to highlight complexity/uniqueness of microbial networks in each land use type.
Answer: We agree with reviewers suggestion of plotting a Venn diagram consisting of keystone taxa between the two land types in the study. However, we did check for any common keystone taxa and have presented the information in Table 3. Accordingly, no shared bacterial and/or fungal keystone taxa were found, and that they were all unique to each land type. Therefore, a Venn diagram cannot be presented or rather we feel would not be informative in the present scenario.
- Line 46: Should the authors clarify that this is for human population? Or do the authors mean the population of any organisms?
Answer: In the previous text the intent was to introduce the importance of ecosystem services to human population because of social-ecological interactions a society has with its ecosystem. Therefore, the sentence is now corrected following to the suggestions of the reviewer.
- Line 58: Please spell out ‘hectares’, otherwise please specify an abbreviation.
Answer: ha is now spelled out as hectares in line 58.
- Line 65-67: In these two references, did they truly conduct experiments to test that these microbes secrete enzymes that contribute to nutrient accumulation? Didn’t they just perform association analyses between presence of microbes and nutrient conditions? If the latter, I do not think these two references provide a strong support for the statement.
Answer: The statement is changed and in line with the supporting literatures.
- Line 67: Abundance of what? Enzyme, nutrient condition, or microbe?
Answer: In the previous sentence the increased abundance was referred to those of the soil bacterial communities in restored loess soils. However, this is now changed in line with the earlier suggestion.
- Line 68ff: Why Chytridiomycota? Other groups of fungi can also efficiently decompose cellulose and chitin such as Agaricomycotina or Pezizomycotina. I am not sure if references 22-24 really support the statement as those are not experimental studies showing the case.
Answer: We agree with the reviewer that members of Agaricomycotina or Pezizomycotina can also efficiently decompose cellulose and chitin. In our study, we found Chytridiomycota to be significantly higher in the AL (1.4%) than in the RL (0.04%) soils (p = 0.009). Therefore, we took the opportune to emphasize on Chytridiomycota fungal phylum in the Introduction section.
The reference 22 was not experimental study and is being delisted. Whereas bibliography 23 and 24 are supportive to the statement and hence retained.
- Line 73 – 74: I am not sure if I understand the sentence correctly. Do the authors mean that as keystone taxa affect many microbial associations so they primarily contribute to changes in soil microenvironment?
Answer: Yes. This understanding is correct. The previous sentence “Keystone microbial taxa that enhance changes in microbial associations are closely related to perturbations in environmental factors” at line 73-74 has been revised to “keystone taxa affect many microbial associations which further drive changes in soil microenvironment along with other physicochemical factors governing the same [25–27].
- Line 79 – 81: Please provide a citation to support this statement. I would recommend rewriting the sentence to something like “Several studies showed that changes in land use affect changes in soil microbial community and their respective keystone taxa.”
Answer: As suggested by the reviewer, the former sentence “Since soil microbial communities and land use are closely associated, any changes in land cover may be reflected by changes in microbial community structure and the integral keystone taxa they harbor”
has been rewritten to ”Several studies showed that changes in land use affect changes in soil microbial community and their respective keystone taxa [29,30].”
- Line 82: Should this be “keystone microbes”? Or rewrite to something like “keystone microbes are significantly increased in vegetation-restored soils …”.
Answer: Line 82 is rewritten as indicated.
- Line 104ff: The authors claim that this microbiome study reflects changes in a “longer-term restoration field experiment”. When I first read this, I expect to see the authors collecting soil data in a time-series fashion to track changes overtime between RL and AL land type. However, I just saw the data collection from one time point. I would recommend the authors to rewrite this sentence to avoid confusing the readers.
Answer: Line 104 is rewritten to indicate the former assessment and longer-term investigations of the effects of anthropogenic interferences in the Chinese Loess Plateau soils and that the data from current study would be a useful resource for future such studies.
- Line 130: I would recommend saying “stored at -20 C prior PCR amplification according to Liu et al. (2018b).”
Answer: Change has been made as indicated.
- Line 140ff: Please clarify whether processing of sequencing reads (like adapter and low-quality base trimming) is required before subsequent analyses.
Answer: Raw sequencing reads with low-quality bases and adapter contamination were trimmed with cut-adapt (v2.3) and Vsearch (v2.13.4_linux_x86_64) software.
- Line 141 – 143: How did the authors normalize the reads? For example, did the authors just randomly pick 49,729 reads in the sample that has larger sequencing reads? Did the authors check whether this trimming process affects the microbial composition?
Answer: Rarefaction method was used to normalize the reads, which randomly selects a certain number of sequences (the smallest or 95% of the smallest sample size) from each sample to reach a uniform sequencing depth to predict the observed OTUs and their relative abundance at threshold sequencing depth (Heck et al., 1975; Kemp and Aller, 2004).
Heck, K.L., van Belle, G., and Simberloff, D. (1975). Explicit Calculation of the Rarefaction Diversity Measurement and the Determination of Sufficient Sample Size. Ecology 56, 1459-1461.
Kemp, P.F., and Aller, J.Y. (2004). Bacterial diversity in aquatic and other environments: what 16S rDNA libraries can tell us. FEMS Microbiol Ecol 47, 161-177.
Further, rarefaction analysis is often used to compare multiple sites at an equal level of effort, and to evaluate changes in community diversity. However, were less interested to check the effects of this trimming process for microbial composition and hence have not conducted the same in our study.
- Line 144: Did the authors have a criterion for ‘high quality sequences’? Please specify.
Answer: ‘high quality sequences’ were obtained after cutting the primers of the sequence, discarding the sequences of unmatched primers, sequence assembly, removing repeated sequences, clustering sequences at 98% similarity level, removing chimeric and singletons OTUs.
Therefore, we have changed “High quality sequences” to “High quality sequences (after sequence filtering, denoising, merging, and the removal of chimeric and singleton)”.
- Line 200: Please provide a citation for the vegan package.
Answer: We have added the following citation for the vegan.
Oksanen, J.; Blanchet, F.G.; Kindt, R.; Legendre, P.; Minchin, P.R.; O’Hara, R.B. Package vegan. R Packag ver 2013.
- Line 275: I do not see Figure S1 showing Chao1 and Shannon Simpson diversity in the manuscript package. Please provide it a supplementary material.
Answer: Figure S1 showing Chao1 and Shannon Simpson diversity has been included now.
- Line 326ff: It would be great if the authors provide a Zi-Pi plot (as a supplementary figure) to illustrate the occurrence of keystone taxa.
Answer: Thanks for the suggestion. A Zi-Pi plot is plotted(see the word file "Answers to Reviewer 2's comment"), however, most of keystone taxa can be seen overlapping in this plot, and poorly illustrate the occurrence of keystone taxa.
Therefore, we kept Table S6 as a clear way to illustrate comprehensive information for keystone taxa. We believe this should suffice the need instead of again showing the same concept via Zi-Pi plot. However, for the sake of convenience and only to show the information to answer this comment from the reviewer, we have separately shown the Zi-Pi plot in the answer to the comments of Reviewer 2.
- Line 374: Should it be ‘confounded’?
Answer: “compounded’ is replaced by ‘confounded’ in line 374.
- Line 369ff: This is hard to judge as the authors do not provide the alpha diversity data (Figure S1) in this manuscript. Probably the authors can help the readers visualize the data by making a plot showing that there is no relationship between soil pH and alpha diversity for each site.
Answer: We have added a new Figure S1 showing no significant microbial diversity differences between AL and RL lands.
- Line 397 – 402: Again, Ascomycota and Basidiomycota are effective cellulose and lignin degraders. I am not sure that this explains why chytrids are more abundant in AL soil. RL soils also have cellulose biomass for microbiome community. Would it be possible that AL soils have more soil moisture content (due to watering) than RL soils so that AL soils attract higher chytrid abundance? Chytrids are often referred as ‘water mold’ so they prefer high moisture for their growth. The authors should have checked the moisture content between AL and RL soil.
Answer: We agree with the reviewer that Ascomycota and Basidiomycota are effective cellulose and lignin degraders. However, the relative abundances of Ascomycota and Basidiomycota, showed no significant differences between AL and RL. As shown in the Figure 2, the relative abundance of Ascomycota was high, but the relative abundance of Basidiomycota was lower, in AL than in the RL. Further, these changes were hard to explain and also why chytrids are more abundant in AL soil.
As suggested by the reviewer, we have checked and added the moisture content in Table S1. However, there was no significant difference between AL (14.0 % ± 3.63) and RL soil (14.7% ± 3.14).
- Line 405ff: I am not sure if this statement is strong unless the authors have the data showing that Sordariomycetes taxa existing in the AL soils represent a diverse array of nutritional modes (plant pathogens, saprobes, epiphytes, coprophiles and fungicoles). Seeing a higher abundance of only ‘Sordariomycetes’, but not at genus or species levels, does not mean that it represents a diverse functionality in the ecosystem.
Answer: We thank the reviewer for improving our understanding and more precise interpretation of the data. This is worth a potential feature in future investigations.
Agreeing to the reviewer’s comment the statement at line 405ff “Such changes can potentially be regarded as an early indicator for resilient communities because many important plant pathogens, saprobes, epiphytes, coprophiles and fungicoles are integral members of the fungal class Sordariomycetes [69,70].” is removed.
- Line 416ff: Could the authors demonstrate this with their existing data? That would make a stronger support.
Answer: “This is further supported by the observation that complex network co-ordination with-in different taxa can compensate the loss of specialist taxa capable of performing particular functions in the soil matrix [71].” has been revised to
“In the previous study, the higher complexity of microbial network was shown to be more resilient to environmental perturbations than simple networks consisting of lower connectivity [78]. For instance, a comparative study of root microbiota in different farming systems found that the microbial network of organic farming was more complex featuring by more highly connected nodes (microbial OTUs) whereas conventional farming network was predominated by weakly connected peripheral nodes [79].”.
- Line 434 – 435: Could the authors make a summarized table showing this ‘land-use specific taxa’? That would be very useful for the readers.
Answer: As suggested by the reviewer, we conducted biomarker species analysis and made a summarized table (Table S7) to show those ‘land-use specific taxa’.
Linear discriminant analysis (LDA) identified 9 bacterial and 151 fungal land-use specific taxa. In the AL soils, Iamia, Marinactinospora, Pyrenochaetopsis, Kazachstania, and Podospora; and in the RL soils, Chthoniobacter, Cenococcum, Tulostoma, and Fabrella were high occurring biomarkers with the highest LDA scores (Table S7).
- Line 437: I think the authors should provide more details on study sites elsewhere in the manuscript (maybe in Materials and Methods). For example, how long have those AL, RL sties been existed? How have they been treated? Providing this info would explain the ‘long term restoration field’ aspect of the current study.
Answer: I order to explain the “long term restoration field” aspect of the current study, we provided the following information in MATERIAL and METHODS section.
“In the northern part, the Grain for Green (GFG) initiative led to the development of various land-use patterns. For instance, in the Ziwuling forest belt of the Ansai county and the Liandaowan town, many previously uncultivable hilly slope farmlands are being revegetated as grasslands, shrub lands, and forest lands [40–43], forming a cluster of rehabilitated lands (RL). Consequently, immense improvement in the soil quality was noted in these RL clusters. In the southern part of the Loess plateau such as the Weibei upland, Guanzhong plain and Luochuan tableland, wide-spread agricul-tural lands (AL) (34°18′–35°50′N; Table S1) prominently grow Malus pumila, Zea mays, Amygdalus persica and Pyrus spp. [44,45]. Large areas of M. pumila have been cultivated for a period of approximately twenty years with high crop turnover. Beside M. pumila, A. persica and P. spp., were the other two local economically important plants [44,45]. Z. mays is usually cultivated as the main crop. The climate is arid and semi-arid with low precipitation and at the agriculturally used soils facing high evaporation rates. These soils are moderate saline-alkali and characterized by a high pH (7.7–8.5) and salt content (averaged 514; Table S1). The accustomed fertilization practice of local farmers include the application of organic fertilizer and high-efficiency compound fertilizer (three nutrients compound N, P, K) fertilizer [46–48]. In compliance with the GFG (year 1999 to 2019) in the Loess hilly regions [41], large areas of slopes or hilly farmlands were re-vegetated to form rehabilitated land (RL) areas since 1999, and growing tree species such as Pinus tabuliformis, Betula platyphylla, Artemisia gmelinii, Caragana Korshinskii, Robinia pseudoacacia, Bothriochloa spp., Artemisia scoparia, Stipa grandis.”
- Line 463 – 464: Is it possible that the finding shows an indicator species for this locality, this climate type or something else?
Answer: A biomarker species analysis was conducted and results summarized in Table S7 showing indicator species of the locality.
- Line 473: Please define how the authors assign Mortierella as generalists based on their data. Please also italicize the genus name Mortierella.
Answer: By generalist we meant to say common presence/persistence in both the AL and the RL soils. Therefore, “the Mortierella (belonging to the order of Mortierellales) are generalists in both soils.” is revised to “the Mortierella (belonging to the order of Mortierellales) act as module connectors (with among-module connectivity (Pi) > 0.62) and present in both the AL and the RL soils.”
- Line 480ff: I am skeptical how the authors make a conclusion here. The authors mention only keystone taxa in RL, but not in AL. Then, the authors cite other studies that found AMF in AL and conclude that keystone taxa in RL are different from AL. I think the authors should have used data from their study for comparison and inference.
Answer: We are in high agreement with the reviewer’s opinion. The new paragraph has been added to make comparison and inference, as shown below:
“In our studied AL soils, there were only two identified fungal keystone taxa, one was unidentified Microascaceae (OTU 13053), and the members from family Microascaceae, which mostly consist of saprobes and plant-pathogenic genera include, Microascus and Pseudallescheria [98]. Another fungal keystone was Talaromyces marneffei (OTU 5193) from genus Talaromyces. T. marneffei is the known dimorphic species producing filamentous growth and yeast phase at different temperature, indicating its strong reproductive strategy. Moreover, T. marneffei is an emerging fungal pathogen causing mycosis in immune-compromised East Asian population [99] whose occurrence in the Loess barren agricultural soils in an early indication of probable health risk factor from these soils. On the other hand, under organic farming practices, Banerjee, Walder, et al., (2018) reported majority of key-stone taxa belonging to arbusuclar mycorrhizal fungi (AMF) and further to fungal orders that can form neutral and beneficial interactions with plants.”
- Line 494 – 495: The avgK values from Table 1 show that RL soils have more disturbance than AL soils, which seems counterintuitive with this statement.
Answer: higher avgK and or avgCC values might indicate higher sensitivity of a microbial community structure to external variations [1,2]. Based on network topological values, we can describe what will happen in a microbial network to potential disturbance, but not what had been happened.
1. Lee, S.A.; Kim, J.M.; Kim, Y.; Joa, J.H.; Kang, S.S.; Ahn, J.H.; Kim, M.; Song, J.; Weon, H.Y. Different types of agricultural land use drive distinct soil bacterial communities. Sci. Rep. 2020, 10, 1–12, doi:10.1038/s41598-020-74193-8.
2. Cheng, X.; Yun, Y.; Wang, H.; Ma, L.; Tian, W.; Man, B.; Liu, C. Contrasting bacterial communities and their assembly processes in karst soils under different land use. Sci. Total Environ. 2021, 751, 142263, doi:10.1016/j.scitotenv.2020.142263.
Therefore, to make our statements clearer,
We changed “the soil bacterial and fungal networks tended to be more disturbed in the RL than in the AL soils” to “the soil bacterial and fungal community structures tend to be more susceptible to potential disturbances in the RL than in the AL soils.” in subsection 3.2.
In the CONCLUSION section, we revised “in the agricultural soils, the land management practices caused disturbances in the soil environment” to “in agricultural ecosystem, soils are more prone to the artificial management practices leading to increased disturbance in soil micro-environment”.
- Table 3 and S6: It looks like Table 3 presents a subset of keystone taxa from Table S6. Please clarify the difference between Table 3 and Table S6. Probably the authors subset only taxa with taxonomic assignment up to the family level for Table 3?
Answer:Yes, it is correct. Table 3 is the subset of Table S6 but with most possible taxonomic assignment for the identified keystone taxa i.e. upto the genus level in terms of bacterial and species level in fungal taxa. More interested readers can avail the detailed information from Table S6.
- Table S1: Please provide the data for soil moisture content in the table.
Answer: We have included the soil moisture content in Table S1.
- Table S4: NMDS is non-metric score. I think it would be more suitable if the authors use Spearmann’s rank correlation as a non-parametric test, instead of Pearson’s correlation.
Answer: We appreciate reviewer’s concerns and therefore feel the need to explain the followed statistical approach in a clearer way:
The dissimilarity of microbial community (beta-diversity) was calculated using pairwise UniFrac distances between samples. And the Pairwise UniFrac distance were visualized using non-metric multidimensional scaling (NMDS) scores via Vegan in R program. Then, we conducted Pearson correlation analysis between microbial community structure (as indicated by NMDS scores) and soil properties.
This is a very regularly used method for the evaluation of relationship between the microbial community structure matrix and soil parameters data.
See the Table 3 in Liu et al. 2019 (doi.org/10.1016/j.scitotenv.2019.01.097); and the Table 3 in Liu et al. 2018 (doi:10.3389/fmicb.2018.02456).
- Table S6: Please indicate that ‘B-‘ stands for bacterial taxa and ‘F-‘ stands for fungal taxa.
Answer: We have added the footnote stating B- stands for bacterial, and F- stands for fungal taxa in the Table S3.

Round 2
Reviewer 2 Report
Comments for authors (round 2)
I thank the authors for considering my comments to improve the manuscript. Most responses are elaborated and well addressed. Please see below for final minor comments:
- About testing the effects of altitudes and types of plant through the Mantel’s tests, I think it will be good if the authors include these results somewhere in the manuscript (maybe in Table 2). This is to help readers realize that the authors have ruled out other possibilities and that the land use is a primary factor that affects the microbial communities.
- Please indicate in Table 3 caption that keystone taxa in Table 3 is the subset of Table S6 but with most possible taxonomic assignment for the identified keystone taxa i.e. up to the genus level in terms of bacterial and species level in fungal taxa, and that the full list of keystone taxa from the analyses can be found in Table S6.
- Please add details for biomarker species analysis/linear discriminant analysis (LDA) in the Materials and Methods section.
- Do the authors intend to provide Figure S1 as a main figure or a supplementary figure? If as a main figure, please renumber the figure accordingly. If as a supplementary figure, please provide it in a supplementary file separated from the main manuscript.
- Please provide proper citations for cutadapt and Vsearch packages.
- Line 213ff: Please specify that the rarefaction method was used for normalization.
- Line 215: For high quality sequences, do the authors have a minimal cutoff for a sequence length after sequence filtering, denoising, merging, and the removal of chimeric and singleton? If so, please specify in the manuscript.

Author Response
I thank the authors for considering my comments to improve the manuscript. Most responses are elaborated and well addressed. Please see below for final minor comments:
- About testing the effects of altitudes and types of plant through the Mantel’s tests, I think it will be good if the authors include these results somewhere in the manuscript (maybe in Table 2). This is to help readers realize that the authors have ruled out other possibilities and that the land use is a primary factor that affects the microbial communities.
Answer:As suggested by the reviewer, we have included the Mantel’s tests in Table 2.
- Please indicate in Table 3 caption that keystone taxa in Table 3 is the subset of Table S6 but with most possible taxonomic assignment for the identified keystone taxa i.e. up to the genus level in terms of bacterial and species level in fungal taxa, and that the full list of keystone taxa from the analyses can be found in Table S6.
Answer:The suggested explaining sentences have been added in Table 3 caption.
- Please add details for biomarker species analysis/linear discriminant analysis (LDA) in the Materials and Methods section.
Answer:We added “Linear discriminant analysis (LDA) effect size was used to investigate microbial biomarker across the studied agricultural and rehabilitated soils. The most obvious biomarkers were selected based on a threshold of LDA score > 2.0 and p-value < 0.05. ” in the Materials and Methods section.
- Do the authors intend to provide Figure S1 as a main figure or a supplementary figure? If as a main figure, please renumber the figure accordingly. If as a supplementary figure, please provide it in a supplementary file separated from the main manuscript.
Answer:We have removed Figure S1 from the main text, and showed it as a supplementary Figure 1.
- Please provide proper citations for cutadapt and Vsearch packages.
Answer:Two citations have been added.
Martin, M. Cutadapt removes adapter sequences from high-throughput sequencing reads. Embnet J. 2011, 17, 10–12, doi:10.14806/ej.17.1.200.
Rognes, T.; Flouri, T.; Nichols, B.; Quince, C.; Mahé, F. VSEARCH: A versatile open source tool for metagenomics. PeerJ 2016, 2016, 1–22, doi:10.7717/peerj.2584.
- Line 213ff: Please specify that the rarefaction method was used for normalization.
Answer:We have specified the rarefaction method in subsection 3.2
“Rarefaction method was used to normalize the reads, which randomly selects the smallest sample size (49,729 and 60,856 for bacterial 16S rRNA and fungal ITS, respec-tively) from each sample to reach a uniform sequencing depth to predict the observed OTUs and their relative abundance at threshold sequencing depth [54].”
- Line 215: For high quality sequences, do the authors have a minimal cutoff for a sequence length after sequence filtering, denoising, merging, and the removal of chimeric and singleton? If so, please specify in the manuscript.
Answer:We do not have a minimal cutoff for a sequence length.